# Proanthocyanidins Restore the Metabolic Diurnal Rhythm of Subcutaneous White Adipose Tissue According to Time-Of-Day Consumption

**DOI:** 10.3390/nu14112246

**Published:** 2022-05-27

**Authors:** Marina Colom-Pellicer, Romina M. Rodríguez, Jorge R. Soliz-Rueda, Leonardo Vinícius Monteiro de Assis, Èlia Navarro-Masip, Sergio Quesada-Vázquez, Xavier Escoté, Henrik Oster, Miquel Mulero, Gerard Aragonès

**Affiliations:** 1Nutrigenomics Research Group, Department of Biochemistry and Biotechnology, Universitat Rovira i Virgili, 43007 Tarragona, Spain; marina.colom@urv.cat (M.C.-P.); rominamariel.rodriguez@urv.cat (R.M.R.); jorgericardo.soliz@urv.cat (J.R.S.-R.); elia.navarro@urv.cat (È.N.-M.); miquel.mulero@urv.cat (M.M.); 2Center of Brain, Behavior and Metabolism, Institute of Neurobiology, University of Lübeck, Marie Curie Street, 23562 Lübeck, Germany; leonardo.deassis@uni-luebeck.de (L.V.M.d.A.); henrik.oster@uni-luebeck.de (H.O.); 3Unitat de Nutrició i Salut, Centre Tecnològic de Catalunya, Eurecat, 43204 Reus, Spain; sergio.quesada@eurecat.org (S.Q.-V.); xavier.escote@eurecat.org (X.E.)

**Keywords:** acrophase, circacompare, chronobiology, chrononutrition, flavonoids, metabolomics, zeitgeber

## Abstract

Consumption of grape seed proanthocyanidin extract (GSPE) has beneficial effects on the functionality of white adipose tissue (WAT). However, although WAT metabolism shows a clear diurnal rhythm, whether GSPE consumption could affect WAT rhythmicity in a time-dependent manner has not been studied. Ninety-six male Fischer rats were fed standard (STD, two groups) or cafeteria (CAF, four groups) diet for 9 weeks (*n* = 16 each group). From week 6 on, CAF diet animals were supplemented with vehicle or 25 mg GSPE/kg of body weight either at the beginning of the light/rest phase (ZT0) or at the beginning of the dark/active phase (ZT12). The two STD groups were also supplemented with vehicle at ZT0 or ZT12. In week 9, animals were sacrificed at 6 h intervals (*n* = 4) to analyze the diurnal rhythms of subcutaneous WAT metabolites by nuclear magnetic resonance spectrometry. A total of 45 metabolites were detected, 19 of which presented diurnal rhythms in the STD groups. Although most metabolites became arrhythmic under CAF diet, GSPE consumption at ZT12, but not at ZT0, restored the rhythmicity of 12 metabolites including compounds involved in alanine, aspartate, and glutamate metabolism. These results demonstrate that timed GSPE supplementation may restore, at least partially, the functional dynamics of WAT when it is consumed at the beginning of the active phase. This study opens an innovative strategy for time-dependent polyphenol treatment in obesity and metabolic diseases.

## 1. Introduction

Most species have developed circadian clocks in order to anticipate recurring and, thus, predictable environmental changes associated with the 24 h day/night cycle. In mammals, the main synchronizer for these clocks is light, although food intake, sleep/wake, and body temperature cycles also help to maintain an organism’s proper timekeeping. Circadian clocks regulate the expression of many metabolic genes across the day and clock disruption increases the risk of metabolic disorders [1,2,3]. The central regulator of circadian clocks in mammals is located in the suprachiasmatic nucleus (SCN) of the hypothalamus, which orchestrates subordinate clocks in peripheral tissues including white adipose tissue (WAT) [4]. SCN-controlled cues such as hormones and food intake together regulate important functions of WAT including adipocyte differentiation, lipid metabolism, and adipokine expression. Adipokines, in turn, can modulate circadian appetite and energy metabolism rhythms in the brain. Such circadian adipocyte–brain crosstalk plays a crucial role in energy homeostasis [4].

Hypercaloric diets disrupt circadian rhythms in a tissue-specific manner [5]. In addition to food composition, the timing of food intake has a strong impact on circadian WAT homeostasis. When nocturnal rodents have access to food only during the light phase, they gain more weight compared with dark phase-fed animals [6,7]. In fact, light phase-fed animals present phase shifts in lipogenic gene expression rhythms, loss of rhythmicity in lipolytic gene expression in WAT, as well as alterations in leptin and insulin circadian profiles and body temperature rhythms compared with animals with access to food only during the dark phase [6,8,9]. In contrast, dark phase-fed animals exhibit lower circulating leptin and WAT pro-inflammatory cytokine levels than animals eating during the light phase [10].

Moreover, metabolite rhythms in blood are boosted by time-restricted feeding, even when endogenous clocks are disrupted such as in liver-specific *Bmal1* knock-out mice [11]. Interestingly, these rhythmic metabolites are mostly amino acids with the highest expression levels right after feeding, suggesting that the timing of food intake determines amino acid daily rhythms [11]. In this sense, amino acid metabolism could play a crucial role in the pathophysiology of metabolic abnormalities associated with obesity. Observational studies show associations between diurnal variations in amino acid plasma concentrations (including branched-chain amino acids (BCAA), aromatic amino acids, alanine, and glutamine) and insulin resistance and diabetes [12,13]. In fact, under insulin resistance and non-alcoholic fatty liver disease, BCAA oxidation enzymes are downregulated in WAT, but not in skeletal muscle, demonstrating the impact of WAT metabolism on BCAA circulating levels [14,15,16].

Proanthocyanidins are a class of polyphenolic compounds that are attracting considerable interest in the nutraceutical field due to their potential health benefits. They are ubiquitous and present as the second most abundant natural phenolic after lignin. Structurally, proanthocyanidins are phenolic compounds that belong to the class of flavonoids and are oligomers of monomeric (epi)catechin units that can be differentiated into A-type or B-type depending on their interflavanic linkages [17,18]. Specifically, previous studies demonstrated that grape seed proanthocyanidins, which are B-type, have beneficial effects on both WAT physiology and WAT clock regulation [19,20]. This prompted us to ask whether consumption of a grape seed extract enriched in proanthocyanidins (GSPE) could affect WAT rhythmicity in a time-dependent manner. To test this, we evaluated the effect of GSPE consumption on rhythmic metabolites of subcutaneous WAT by administration either at the beginning of the light/rest phase or at the beginning of the dark/active phase in rats with diet-induced obesity.

## 2. Materials and Methods

### 2.1. Proanthocyanidin Extract

The grape seed proanthocyanidin extract (GSPE) used in this study was composed of monomers (21.3%), dimers (17.4%), trimers (16.3%), tetramers (13.3%), and oligomers (5–13 units; 31.7%) of proanthocyanidins according to the manufacturer (Les Dérivés Résiniques et Terpéniques, Dax, France). The phenolic composition of this extract was further analyzed by Margalef et al. [21].

### 2.2. Study Design and Dosage Information

Ninety-six 12-week old male Fischer 344 rats were purchased from Charles River Laboratories (Barcelona, Spain). Animals were housed in pairs under a 12 h:12 h light:dark cycle, at 22 °C, 55% of humidity and fed ad libitum with a standard chow diet (STD) (Panlab A04, Barcelona, Spain) and tap water for one week of adaptation. Then, animals were randomly divided into two groups according to their diet. Thirty-two rats were fed with STD and 64 rats with cafeteria diet (CAF) for 9 weeks. The composition of STD was 76% carbohydrates, 20% protein, and 4% fat. CAF consisted of biscuits with cheese and paté, bacon, ensaïmada (sweetened pastry), carrots, and milk with sucrose 20% (*w*/*v*), and its composition was 51% carbohydrates, 35% fat, and 14% proteins. The treatment period started in week 5 and continued for 4 weeks. STD-fed rats received 450 µL of vehicle at the beginning of the light phase (*zeitgeber* time 0; ZT0; *n* = 16) or at the beginning of the dark phase (ZT12; *n* = 16). CAF animals were divided into four groups (*n* = 16 each) receiving either vehicle or 25 mg of GSPE/kg of body weight at either ZT0 or ZT12. GSPE was dissolved in 450 μL of commercial sweetened skim condensed milk (Nestle; 100 g: 8.9 g protein, 0.4 g fat, 60.5 g carbohydrates, 1175 kJ). Vehicle groups were supplemented with the same volume of sweetened skim condensed milk. Two or three days before administration, rats were trained to voluntarily lick the milk to avoid oral gavaging. Treatment was orally administered daily using a syringe.

Body weight and food intake were recorded weekly during the whole experiment. At the end of the experiment, each group of 16 animals was randomly divided into 4 sub-groups of 4 rats in order to sacrifice them at four different time points across the day at ZT1 (9 a.m.), ZT7 (3 p.m.), ZT13 (9 p.m.) and ZT19 (3 a.m.). Prior to the sacrifice, animals were fasted for 3 h and sacrificed by decapitation. Trunk blood was collected and centrifuged (2000× *g*, 15 min, 4 °C) to obtain serum. In addition, adipose tissue depots were excised and weighed; inguinal WAT fat pads (iWAT) were snap-frozen in liquid nitrogen. Serum and adipose samples were stored at −80 °C until further use.

Animal experiments were approved by the Animal Ethics Committee of Universitat Rovira i Virgili (reference number 9495) and carried out in accordance with Directive 86/609/CEE of the Council of the European Union and the procedures established by the Departament d’Agricultura, Ramaderia i Pesca of Generalitat de Catalunya (Barcelona, Spain).

### 2.3. Biometric Parameters and Circulating Biomarkers

Body weight gain was calculated by comparing body weight at the beginning of the treatment period and at the end of the experiment. Body fat content was determined by summarizing the weight of all fat pads. Relative weights of iWAT were calculated dividing fat pad weight by total body weight. Food intake was determined weekly by calculating the weight difference between food prior to placement in the cage and food leftovers after 24 h. Energy intake was calculated according to the caloric content of each diet provided by the manufacturer. Accumulated food intake represents all the calories eaten during the treatment period. Enzymatic colorimetric assays were used for the determination of serum glucose, triglycerides, total cholesterol (QCA, Barcelona, Spain), and non-esterified fatty acids (Wako, Neuss, Germany).

### 2.4. Adipose Tissue Histology

This study focused on inguinal subcutaneous WAT (iWAT). Rats sacrificed at ZT1 and ZT7 (*n* = 4 each) in the six different groups were used for the histological analysis. Small pieces of frozen iWAT were thawed and fixed in 4% formaldehyde. Paraffin embedding and sectioning, hematoxylin-eosin staining, and calculations for the area, volume, and number of adipocytes were performed following Gibert-Ramos et al. [22]. The distribution of adipocyte sizes across the tissue was calculated by distributing all counted cells of each sample into two groups according to their area (<3000 μm^2^ or >3000 μm^2^); then, the number of total counted adipocytes was used to calculate the percentage of adipocytes in both categories.

### 2.5. Adipose Tissue Preparation for ^1^H NMR-Based Metabolomics Assay

Metabolites from iWAT were analyzed by untargeted nuclear magnetic resonance spectrometry (NMR). Hydrophilic and lipophilic metabolites were extracted from the fat pad following the procedure described by Castro et al. [23]. Two hundred mg of adipose tissue was homogenized with 800 µL of methanol, 1600 µL of chloroform, and 800 µL of Milli-Q water using vortex. Homogenates were centrifuged at 4000× *g* for 10 min at 4 °C in 15 mL Falcon tubes. Supernatants (hydrophilic metabolites) were separated from pellets (lipophilic metabolites). Pellets were washed following the previous procedure. Lipophilic metabolites were separated and dried using a nitrogen stream. Hydrophilic metabolites were frozen overnight at −80 °C and lyophilized. Both hydrophilic and lipophilic phases were stored at −80 °C.

### 2.6. NMR Analysis

NMR measurements of hydrophilic and lipophilic extracts were performed following the protocol of Palacios-Jordan [24]. For metabolite identification, the acquired ^1^H NMR spectra were compared to references of pure compounds from the metabolic profiling AMIX spectra database (Bruker), HMDB, Chenomx NMR suite 8.4 software (Chenomx Inc., Edmonton, AN, Canada). Metabolites were assigned by ^1^H–^1^H homonuclear correlation (COSY and TOCSY), ^1^H–^13^C heteronuclear (HSQC) 2D NMR experiments, and by correlation with pure compounds run in-house. After pre-processing, specific ^1^H NMR regions identified in the spectra were integrated using the AMIX 3.9 software package. A data matrix was generated with absolute concentrations derived from, both, lipophilic and hydrophilic extracts. Data were scaled to the same units for all the identified metabolites.

### 2.7. Statistical Analysis

Statistical tests were performed using XL-Stat 2017 software (Addinsoft, Paris, France), and graphics were prepared using GraphPad Prism 9 (GraphPad Software, San Diego, CA, USA). A *p*-value < 0.05 was considered statistically significant. Data were analyzed using one-way analysis of variance (ANOVA) or Kruskal–Wallis test, depending on whether data were parametric or non-parametric (tested by Shapiro–Wilk test), followed by Bonferroni post-hoc test when comparing individual time points between the different treatment groups. Student’s *t*-test or Mann–Whitney tests were used for pairwise comparisons. For correlation analysis, Spearman correlation was performed using the Harrell Miscellaneous package (version 4.6) and significant correlations were classified when *p*-value < 0.05. Corrplot package (version 0.9) was used for visualization of the correlations for each group. Diurnal (i.e., 24 h) rhythmicity was evaluated using a sample number of 4 animals per timepoint on absolute values. Diurnal parameters such as rhythmicity, mesor, amplitude, and acrophase were calculated using CircaCompare algorithm [25]. Rhythmic parameters were compared in a pairwise fashion using CircaCompare. Presence of metabolite rhythmicity was considered when a *p*-value < 0.05 was found. Comparison of amplitudes and mesors was performed by fitting cosine curve regardless of the rhythmicity (*p*-value = 1). This method allows for an estimation of mesor and amplitude between rhythmic and non-rhythmic metabolites. Phase estimation was performed only when both metabolites were considered rhythmic (CircaCompare *p*-value < 0.05). Graphs showing diurnal rhythm and acrophase were created using the Python package based on Cosinor and MetaboAnalyst 5.0 (v11.0, Wishart Research Group, University of Alberta, Edmonton, Canada) was used to analyze metabolic pathway involvement (https://www.metaboanalyst.ca accessed on 18 January 2022) [26].

## 3. Results

### 3.1. GSPE Administration at ZT12 Reduces Body Weight Gain and Accumulated Food Intake

Our experiment confirmed that cafeteria diet is a robust model of diet-induced obesity in rats. After 9 weeks of obesogenic diet, final body weight and body weight gain of CAF animals were significantly higher compared with STD animals (Table 1). STD-VH (STD supplemented with vehicle) and CAF-GSPE animals supplemented at night (ZT12) presented lower final body weight, and lower body weight gain in CAF-GSPE, compared with the respective groups supplemented in the morning (ZT0). Interestingly, when GSPE was administrated at night, body weight gain was as low as in the STD-VH group and different from CAF-VH animals. Moreover, body fat content was significantly higher in CAF rats compared with STD-fed rats in both ZT0 and ZT12 animals. However, iWAT relative weight in CAF-VH supplemented at night was not statistically different compared to STD or CAF-GSPE. As expected, food intake was higher in CAF groups irrespective of the time of supplementation. CAF-GSPE supplemented at night presented lower accumulated food intake compared with CAF-GSPE supplemented in the morning. Interestingly, CAF-GSPE rats presented lower final food intake compared with CAF-VH only at ZT12 supplementation.

### 3.2. Adipocyte Size Is Modified According to Time-of-Day GSPE Administration

Subcutaneous adipose tissue expansion is associated with metabolic health. In fact, in rodent models, surgical removal of iWAT can eventually lead to metabolic dysfunction and ectopic lipid accumulation [27,28]. Conversely, transplantation of iWAT into the visceral cavity of mice leads to improved glucose homeostasis and a reduction in body weight and total fat mass [29]. Therefore, in this study, we also evaluated the effect of GSPE administration on the histology of iWAT. Accordingly, CAF diet consumption resulted in larger adipocytes compared with STD-VH groups (Appendix A). Specifically, CAF animals presented higher adipocyte area and volume, and lower percentage of smaller adipocytes in both ZT0 and Z12 animals compared to STD groups. Importantly, GSPE consumption resulted in a stronger effect on adipocyte histology, lowering the area and volume of adipocytes, when it was supplemented at ZT0 compared with animals supplemented with GSPE at ZT12.

### 3.3. GSPE Administration at ZT12 Recovers the Diurnal Rhythmic Concentration of Serum TAG That Is Lost in Diet-Induced Obesity

The CircaCompare algorithm was used to evaluate the effect of GSPE supplementation on the diurnal rhythmicity of triglyceride (TAG), glucose, cholesterol and non-esterified fatty acid (NEFA) serum concentrations. Rhythm parameters including rhythmicity, mesor, amplitude, and acrophase were calculated and compared in a pairwise fashion (Table 2). STD animals treated at ZT0 showed significant rhythmicity only in TAG, while CAF animals presented a diurnal rhythm in most of the biochemical parameters. However, in the ZT12 groups, TAG (*p*-value = 0.002) and cholesterol (*p*-value = 0.035) serum concentrations presented a significant rhythmicity in STD conditions that was completely lost in response to CAF diet. Importantly, GSPE administration recovered the diurnal rhythm of TAG concentrations (*p*-value = 0.019) when it was consumed at ZT12. In addition, CAF diet resulted in higher TAG and glucose mesor concentrations compared with the respective STD groups. However, no effect of GSPE was observed in these parameters. In contrast, cholesterol mesor concentrations and TAG amplitude estimations were significantly increased in response to CAF diet only in ZT0 treated animals and GSPE consumption restored cholesterol levels to STD conditions only when it was consumed at ZT0.

### 3.4. Metabolite Concentrations in iWAT Are Restored in Response to GSPE Administration in a Time-Dependent Manner

Next, ^1^H NMR was performed to determine whether the metabolite profile of iWAT in response to GSPE was significantly influenced by the time-of-day of its consumption. After alignment and normalization of the spectra, a total of 45 metabolites were identified and integrated, 10 in the lipid phase and 35 in the aqueous phase. Table 3 and Table 4 show the 24 h mean concentrations of all metabolites detected in this tissue.

In the lipid phase (Table 3), CAF diet induced changes in the 24 h mean concentrations of omega-3, oleic acid, linoleic acid, PUFAs, and total fatty acids compared with STD-VH and, when GSPE was administered, only the levels of total fatty acids decreased partially to basal levels in both ZT0 and ZT12 groups. Interestingly, a similar pattern was observed for TAGs, but their concentrations only decreased in response to GSPE when it was administered at ZT12.

In contrast, in the aqueous phase (Table 4), CAF diet decreased the 24-h mean concentrations of aspartate, lysine, carnitine, taurine, propionate, glycerophosphocholine, choline, inosine, histamine, and glucose compared to STD-VH in the ZT0 group but not at ZT12. In addition, in ZT0 animals, GSPE administration increased the concentrations of lysine, propionate, glycerophosphocholine, choline, inosine, histamine, and glucose. Similarly, when GSPE was administrated at ZT12, propionate concentrations achieved values not significantly different from STD-VH.

To further evaluate the effect of GSPE administration on the global functionality of iWAT, we also performed a global correlation analysis between the 24 h mean concentrations of all detected metabolites and the biometric parameters collected in the last week of the experiment (Appendix A). Collectively, CAF diet resulted in a clear reduction in the number of correlations (26.2% lower in the ZT0 and 22.1% lower in the ZT12 groups) while GSPE supplementation restored the correlation network to similar levels as in the STD group for both ZTs.

### 3.5. GSPE Administration at ZT12 Re-Establishes the Number of Rhythmic Metabolites in iWAT

To specifically evaluate the diurnal rhythm of detected iWAT metabolites, each group of animals was divided into four subgroups according to time of sacrifice (ZT1, ZT7, ZT13, or ZT19). The CircaCompare algorithm was used to model the oscillation profiles of these metabolites across 24 h, and rhythm parameters including rhythmicity, mesor, amplitude, and acrophase were calculated and compared in a pairwise fashion (Appendix A).

As summarized in Figure 1a, STD animals at ZT0 presented four metabolites (glutamate, asparagine, uracil, and valine) with diurnal rhythmic concentrations, and only valine concentrations preserved rhythmicity in the CAF group. Moreover, leucine, isoleucine, glycogen, glycerol, tyrosine, and fumarate also presented rhythmic concentrations in this group of animals. Interestingly, the supplementation of GSPE resulted in loss of seven rhythmic metabolites found in CAF animals, and only concentrations of 3-hydroxybutyrate (3-OHB) became rhythmic in response to GSPE administration. In contrast, the STD animals treated at ZT12 presented 18 metabolites with rhythmic concentrations. However, under CAF diet, 16 metabolites lost rhythmicity and only fumarate and lactate concentrations kept stable diurnal rhythms in this group of animals. Importantly, GSPE supplementation at ZT12 restored the rhythm of 12 of 16 metabolites. In total, GSPE administration at ZT12 resulted in 27 rhythmic metabolites (Figure 1b).

Five of these twelve rhythmic metabolites that were restored by GSPE administration at ZT12 (oleic acid, linoleic acid, total fatty acids, triglycerides, and mono-unsaturated fatty acids (MUFAs)) were detected in the lipid phase (Figure 2). MUFAs did not show any difference in the rhythmic parameters comparing CAF-GSPE and STD-VH animals. However, oleic acid, linoleic acid, total fatty acids, and triglycerides concentrations differed on mesor value, being higher in all of them except for linoleic acid, which was lower in CAF-GSPE compared with STD-VH. Moreover, GSPE supplementation also resulted in a phase delay of 5 h in oleic acid compared with the STD-VH group. In the aqueous phase, lactate presented rhythmicity in all animal groups at ZT12 and formate was rhythmic in both CAF-VH and CAF-GSPE groups and, interestingly, GSPE caused a phase advance of 9 h. Notably, glutamine, succinate, phosphorylcholine, glycerol, taurine, tyrosine, and 3-OHB restored diurnal rhythmicity only when GSPE was administered at ZT12 (Figure 3). Among these, phase advances of 6, 5, and 3 h were detected in tyrosine, phosphorylcholine, and 3-OHB concentrations, respectively.

In summary, 19 of 45 detected metabolites in iWAT were rhythmic in the STD-VH group and 18 of these 19 metabolites were rhythmic in STD animals treated at ZT12. Only 4 metabolites of these 19 were rhythmic at ZT0. In response to CAF diet, the number of rhythmic metabolites decreased dramatically, however, GSPE recovered their rhythmicity when it was administrated at ZT12.

### 3.6. GSPE Administration at ZT12 Restores the Rhythmicity of Alanine, Aspartate and Glutamate Metabolism Pathways in iWAT

Next, to test whether the identified rhythmic metabolites at ZT12 were biologically meaningful, we performed a pathway analysis using MetaboAnalyst 5.0 software of the 12 hydrophilic rhythmic metabolites detected in STD group as well as the 20 rhythmic hydrophilic metabolites detected in response to GSPE administration at ZT12.

As shown in Table 5, rhythmic metabolites in the STD group were significantly enriched for alanine, aspartate and glutamate metabolism (*p*-value = 1.10 × 10^−^^6^), arginine biosynthesis (*p*-value = 1.30 × 10^−^^4^), and D-glutamine and D-glutamate metabolism (*p*-value = 8.50 × 10^−^^4^). Notably, in animals supplemented with GSPE at ZT12, rhythmic metabolites were enriched for alanine, aspartate, and glutamate metabolism (*p*-value = 3.80 × 10^−^^4^) suggesting that GSPE administration at ZT12 restored the rhythmicity of this metabolic pathway in iWAT that was lost in response to CAF diet. In fact, succinate and glutamine, the most important metabolites involved in the alanine, aspartate, and glutamate metabolism pathway, were rhythmic in both STD-VH-ZT12 and CAF-GSPE-ZT12 groups (Figure 4a,b). In addition, rhythmic hydrophilic metabolites of animals supplemented with GSPE at ZT12 also presented an enrichment for the phenylalanine, tyrosine, and tryptophan biosynthetic pathway (*p*-value = 9.90 × 10^−^^4^), suggesting that the 13 rhythmic hydrophilic metabolites exclusively detected in animals supplemented with GSPE, and not in the STD group, could be associated with this biosynthetic pathway.

## 4. Discussion

The health effects of proanthocyanidins are usually determined without considering biological rhythms. However, it seems evident that they should be considered in chrononutrition studies as proanthocyanidins can significantly modulate the circadian system. In fact, previous studies demonstrated that GSPE alters the expression profile of clock genes in the rat hypothalamus and modulates the serum melatonin concentrations when it was administered at the beginning of the light phase [30]. In addition, GSPE critically modulates hepatic BMAL1 acetylation, *Nampt* expression, and nicotinamide adenine dinucleotide (NAD) availability in mice as well as the expression of *Clock* and *Per2* genes in white adipose tissue [20,31]. Nevertheless, other mechanisms cannot be discarded because GSPE was also shown to increase endothelial nitric oxide synthase (eNOS), 5′-AMP-activated protein kinase (AMPK), and SIRT1 expression in the light cycle of rats [32]. However, to the best of our knowledge, no study so far has addressed the effect of GSPE consumption on the diurnal rhythmicity of subcutaneous WAT. In this context, we investigated the effect of GSPE administration on the diurnal oscillations of adipose tissue metabolites (assessed by NMR) in rats with diet-induced obesity. Our results provide evidence that grape seed proanthocyanidins, essentially B-type proanthocyanidins, are able to restore the diurnal rhythm of alanine, aspartate, and glutamate metabolic pathway in iWAT when they are administered at ZT12.

### 4.1. Biometric Parameters and Circulating Biomarkers

Although various studies have shown that grape seed proanthocyanidins could prevent body weight gain and fat accumulation, there are scarce data about the effect on body fat content at different ZT points. In this study, although our data did not demonstrate an effect of GSPE on body fat content, we report that the administration of GSPE reduced body weight gain and accumulated food intake when it was consumed at the beginning of the dark phase. In fact, our group recently observed very similar results in Wistar rats supplemented with GSPE at dose of 25 mg/kg of body weight for 4 weeks [33]. Consistent with these previous results, our data also demonstrated that GSPE consumption significantly altered the pattern of expansion of iWAT by decreasing adipocyte size and increasing adipocyte number. In addition, our data showed for the first time that GSPE consumption recovered the diurnal rhythmic concentrations of serum TAG that were lost in diet-induced obesity only when it was administered at ZT12. Glucose and lipid metabolism present circadian regulation to prepare the metabolic functions for the outcoming situations. Although several studies have previously demonstrated that GSPE consumption properly restore blood lipid concentrations in animals with obesity, our study showed that GSPE enhanced the diurnal rhythm of TAG and cholesterol concentrations according to time-of-day consumption. By contrast, in our experiment, both glucose and NEFA serum concentrations did not display diurnal rhythmicity in response to GSPE consumption. However, as we evaluated rhythm variables only at 6 h intervals, we cannot be completely sure that the consumption of GSPE exclusively affected the diurnal rhythm of TAG and cholesterol concentrations. Thus, more studies with a 4 h interval evaluation are needed to elucidate the effects of GSPE consumption on the circadian concentrations of other circulating biomarkers.

### 4.2. Lipophilic Metabolites in Adipose Tissue

As expected, the concentration of detected lipophilic metabolites in iWAT was much higher compared to hydrophilic metabolites. Although we did not observe a significant effect of treatment time on the concentration of lipophilic metabolites in iWAT, CAF diet affected significantly its concentrations. In fact, CAF diet determined the degree of unsaturation of fatty acids resulting in lower concentrations of polyunsaturated fatty acids (omega-3, linoleic acid and PUFAs) and higher concentrations of monounsaturated oleic acid compared with STD diet animals. In addition, it is known that CAF disrupts circadian rhythms [34]. In this sense, our findings support a study demonstrating an increase in TAG containing lower degree of unsaturation of fatty acids in subcutaneous WAT of mice with ﻿adipocyte-specific deletion of the clock gene *Arntl* (also known as *Bmal1*) compared with control mice [23]. BMAL1 is essential for the function of the molecular circadian clock and regulates circadian transcription of metabolic genes. In our study, MUFAs and omega-3, linoleic and oleic fatty acids presented diurnal rhythms in STD-VH and CAF-GSPE groups treated at ZT12, suggesting a metabolically active subcutaneous fat pad when GSPE was supplemented at the beginning of the dark phase. Consistent with these results, Castro et al. [23] also detected clear changes of lipids across light/dark cycle in brown adipose tissue, while in WAT significant differences were not detected. These results were expected for the authors as brown adipose tissue has a higher dynamic metabolic role in comparison with WAT. However, it is known that subcutaneous WAT also may develop brown-like adipocytes. Thus, further research is warranted to elucidate the clinical relevance of these lipidic rhythmic oscillations in response to GSPE consumption in this tissue.

### 4.3. Lipid Metabolism in Adipose Tissue

Lipolysis and lipogenesis are mainly influenced by the light/dark cycle and fasting/feeding states. The light/dark cycle regulates melatonin, which is a hormone that plays a key role in many physiological processes such as lipid metabolism [35]. In our study, total fatty acids, TAGs, lactate, glycerol, and phosphorylcholine presented higher concentrations during the dark phase in STD-VH and CAF-GSPE animals treated at ZT12. The acrophase in the dark phase of these metabolites indicates an activation of lipid synthesis during the dark phase and lipid oxidation during the light phase, respectively [23]. In fact, after feeding, glucose and fatty acids are taken up by WAT to synthesize acetyl-CoA for fatty acid synthesis [36]. Glycerol, another product of glycolysis, is needed for TAG formation in WAT when it is converted into glycerol-3-phosphate [37]. Lactate, for which we also found its acrophase in the dark phase, increases its production in WAT after glucose intake [36,37]. Moreover, we expected to find 3-OHB increased during the light phase because of its association with β-oxidation [23]. However, in our study, this metabolite was increased during the dark phase in STD animals and in response to GSPE administration at ZT12 when lipogenesis is active in this tissue, suggesting that 3-OHB could play an alternative role as a substrate for ATP generation during prolonged catabolic periods [23].

### 4.4. Hydrophilic Metabolites in Adipose Tissue

Whereas the role of adipose tissue in glucose and lipid homeostasis is widely recognized, its role in systemic protein and amino acid metabolism is less well-appreciated. Mammals obtain energy from proteins only in extreme situations such as long-term starvation and, in normal conditions, amino acids are recycled for new proteins. Interestingly, protein-associated metabolic pathways also present circadian rhythms in human subcutaneous WAT with higher activity during the day such as branched-chain amino acid (BCAA) metabolism [38]. In fact, WAT may be the primary tissue which uptakes BCAAs from liver after feeding as insulin action and glucose uptake increase BCAA uptake in WAT [39]. BCAAs can be taken up by WAT through direct BCAA transport or, depending on BCAA levels, inside the adipocyte, exchanging internal glutamine, alanine, and asparagine for external BCAA and glutamate, which occurs immediately after feeding [40]. BCAA uptake and metabolism in WAT are reduced under insulin resistance and inflammation in humans and rodents and, consequently, are associated with elevated blood BCAA levels. A previous study reported a significant reduction on BCAA levels in WAT under a high-fat diet compared to STD [5,39]. Although in our study the reduction of BCAA levels in CAF groups was not significant in iWAT, our findings demonstrated important differences in the diurnal rhythm of amino acids in response to CAF diet as well as in response to GSPE administration at ZT12. Specifically, from the 18 rhythmic metabolites found in STD animals, 16 lost their rhythm under CAF diet, and importantly, most of them restored the rhythmicity only when GSPE was administrated at ZT12. In addition, these metabolites, including alanine, asparagine, aspartate, fumarate, glutamate, glutamine, and succinate, were significantly involved in alanine, aspartate, and glutamate metabolism, suggesting that this metabolic pathway could prevent the metabolic abnormalities associated with an obesogenic diet.

### 4.5. Alanine, Aspartate and Glutamate Metabolism

Glutamine and alanine are involved in a mechanism to avoid nitrogen toxification in WAT. The regeneration of proteins drives to nitrogen removal. In fact, the way to remove nitrogen in WAT and skeletal muscle is through the synthesis of glutamine and alanine. During starvation state, some authors reported significant release of glutamine, alanine, taurine, and tyrosine by subcutaneous WAT probably due to protein breakdown [40,41]. Glutamine can be synthesized and kept in WAT for lipid synthesis or liberated in the circulation to be uptaken for other tissues including kidney and intestine, which consume glutamine as source of energy through the TCA cycle. Thus, WAT plays a significant role in whole-body glutamine homeostasis as it releases 20% of net amounts of glutamine. Alanine can also be synthesized and liberated by WAT. The synthesis and transport of alanine to the liver, where the urea cycle takes place, is a mechanism to ensure the prevention of nitrogen toxification. In the liver, alanine could also be converted to glucose through gluconeogenesis, a substrate for muscle, brain, erythrocytes, and other glucose-dependent cells. A previous study in mice reported that BCAA production in WAT is higher during the light phase, which matches with the timing of alanine production [5]. However, in our study, alanine concentrations were higher during the dark phase in response to GSPE administration at ZT12. Indeed, in these animals, all the metabolites involved in alanine, aspartate, and glutamate metabolism showed an acrophase during the dark phase. In contrast, STD animals presented acrophases of asparagine and glutamate during the light phase with peaks around ZT3, but the administration of GSPE at ZT0 could not restore the rhythmicity of these metabolites under CAF diet conditions. Therefore, timing of supplementation is crucial for affecting the acrophase of alanine, aspartate, and glutamate metabolism, and only GSPE supplementation at ZT12 restored the rhythmicity lost in response to CAF diet. Glutamine and succinate in WAT also attenuate the expression of pro-inflammatory genes and macrophage infiltration [42,43]. Pro-inflammatory genes, such as TNF-α and IL-6, exhibit circadian rhythmicity with the highest concentrations during the light phase when glutamate and succinate concentrations are low in iWAT in response to GSPE administration at ZT12 [44]. In human WAT, oxidoreductase activity was shown to be higher in the evening, at the beginning of the rest phase for humans [38]. ﻿Therefore, the observed diurnal rhythm of glutamine and succinate is in coherence with pro-inflammatory gene expression in STD animals and also in response to GSPE administration at ZT12. However, further studies are needed to analyze the inflammatory status of this tissue and and confirm these results

## 5. Conclusions

Cafeteria diet induced greater accumulation of monounsaturated than polyunsaturated fatty acids in iWAT compared with non-obese animals. Although GSPE supplementation did not present any effect on the degree of (un)saturation of fatty acids, its consumption restored the diurnal rhythm of 12 metabolites only when it was administered at the beginning of the dark/activity phase (ZT12). Diet-induced obesity is associated with metabolic disruption of adipose tissue including diurnal rhythm misalignment. Therefore, the amelioration of diurnal rhythmicity could enhance the functionality of this tissue. Our results demonstrated for the first time that timed GPSE administration specifically at ZT12 restored the rhythmic metabolism of alanine, aspartate, and glutamate that was lost in response to CAF diet. This pathway plays a pivotal role in the metabolism of nitrogen-containing compounds, BCAA metabolism, and pro-inflammatory state in mammals. However, further studies are needed to elucidate the metabolic pathways and processes involved in these events and better understand the interaction of diurnal rhythms and proanthocyanidins on the circadian biology of WAT.

## Figures and Tables

**Figure 1 nutrients-14-02246-f001:**
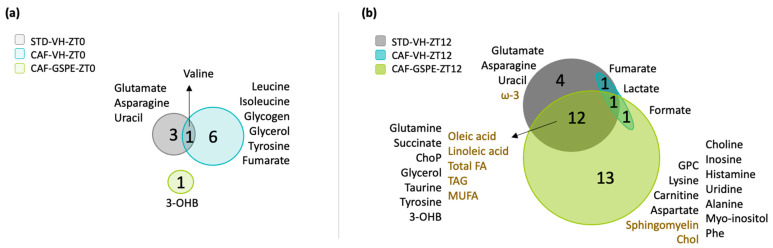
Overlap diagram of rhythmic metabolite concentrations presented in ZT0 (**a**) and ZT12 groups (**b**). Rats were fed standard or cafeteria diet for 9 weeks. At week 6, rats received a daily oral dose of GSPE or vehicle for 4 weeks at ZT0 or ZT12. Rats from each group were sacrificed at four different time points: ZT1, ZT7, ZT13, and ZT19 in order to analyze the diurnal rhythm of these metabolites. Forty-five metabolites detected by NMR were analyzed using CircaCompare algorithm based on Cosinor (*p* < 0.05). Metabolites from aqueous phase are shown in black color and metabolites from lipidic phase in brown color. Abbreviations: iWAT, inguinal adipose tissue; ZT0, beginning of the light phase; ZT12, beginning of the dark phase; STD-VH-ZT0; standard diet vehicle group treated at ZT0; CAF-VH-ZT0, cafeteria diet vehicle group treated at ZT0; CAF-GSPE-ZT0, cafeteria diet treated with grape-seed procyanidin extract at ZT0; STD-VH-ZT12; standard diet vehicle group treated at ZT12; CAF-VH-ZT12, cafeteria diet vehicle group treated at ZT12; CAF-GSPE-ZT12, cafeteria diet treated with grape seed procyanidin extract at ZT12; 3-OHB, 3-Hydroxybutyrate; Chol, cholesterol; Total FA, total fatty acids; TAG, triacylglyceride; MUFA, monounsaturated fatty acids; ω-3, omega-3; GPC, glycerophosphocholine; Phe, phenylalanine; ChoP, phosphorylcholine.

**Figure 2 nutrients-14-02246-f002:**
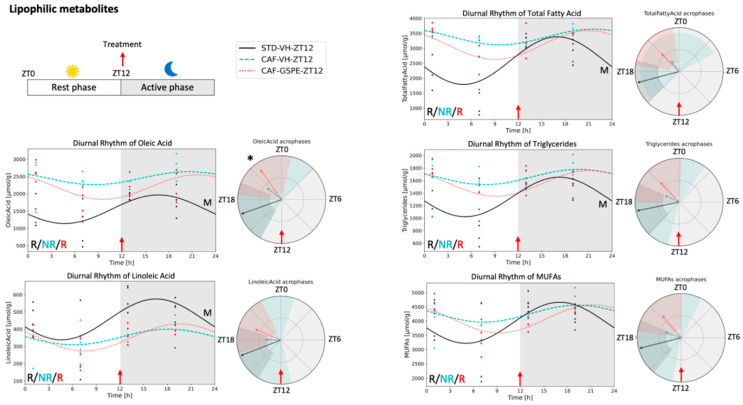
Lipophilic metabolites that recovered their diurnal rhythm loss under cafeteria diet in response to GSPE administration at ZT12. Circadian parameters such as rhythmicity, mesor, amplitude, and acrophase were calculated using the CircaCompare algorithm based on Cosinor. Graphs showing diurnal rhythm and acrophase were performed using the Python package based on Cosinor. R indicates significant rhythmicity; NR indicates non-rhythmic; M indicates significant mesor difference between STD-VH-ZT12 and both cafeteria diet groups; ***** indicates significant acrophase difference between STD-VH-ZT vs CAF-GSPE-ZT12. None of the metabolites presented differences between groups for amplitude.

**Figure 3 nutrients-14-02246-f003:**
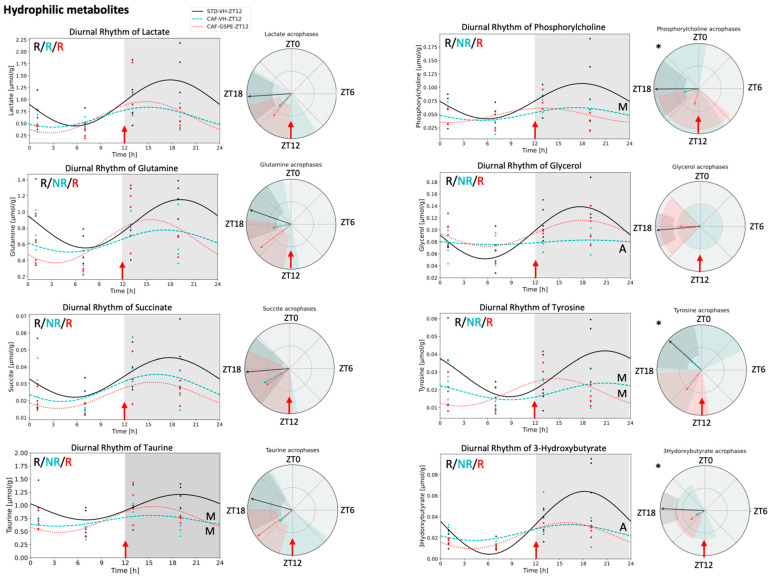
Hydrophilic metabolites which recovered their diurnal rhythm lost under cafeteria diet through GSPE administration in ZT12. Circadian parameters such as rhythmicity, mesor, amplitude, and acrophase were calculated using the CircaCompare algorithm based on Cosinor. Graphs showing diurnal rhythm and acrophase were performed using the Python package based on Cosinor. R, indicates significant rhythmicity; NR, indicates non-rhythmic; M, denotes significant mesor difference against STD-VH-ZT12; A, denotes significant amplitude difference against STD-VH-ZT12; ***** indicates significant acrophase difference between STD-VH-ZT vs CAF-GSPE-ZT12.

**Figure 4 nutrients-14-02246-f004:**
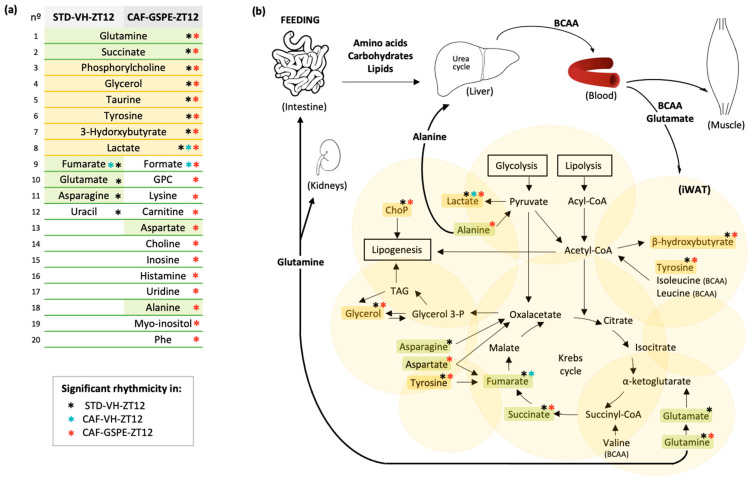
Metabolic network analysis of identified rhythmic metabolites in iWAT. Hydrophilic metabolites with diurnal rhythm in the corresponding groups, (**a**) inter-tissue amino acid flux and alanine, aspartate, and glutamate metabolism pathway (**b**). Green boxes indicate metabolites involved in alanine, aspartate, and glutamate metabolism that are rhythmic in STD-VH-ZT12, CAF-GSPE-ZT12, or both groups; yellow boxes indicate metabolites that are rhythmic in both STD-VH-ZT12 and CAF-GSPE-ZT12 that are not involved in alanine, aspartate, and glutamate metabolism. Abbreviations: ChoP, phosphorylcholine; GPC, glycerophosphocholine; Phe, phenylalanine; TAG, triacylglyceride; Glycerol 3-P, glycerol 3-phosphate.

**Table 1 nutrients-14-02246-t001:** Biometric parameters of rats fed standard or cafeteria diet supplemented with vehicle or GSPE at ZT0 or ZT12 during the last 4 weeks of the experiment.

	ZT0 SDT-VH	ZT0 CAF-VH	ZT0 CAF-GSPE	ZT12 STD-VH	ZT12 CAF-VH	ZT12 CAF-GSPE
Body weight (g)	514.7 ± 7.5 ^a^	591.1 ± 7.8 ^b^	573.1 ± 9.2 ^b^	481.8 ± 7.1 ^a^*	569.3 ± 8.4 ^b^	546.8 ± 7.8 ^b^*
Body weight gain (g)	47.3 ± 3.4 ^a^	74.4 ± 3.9 ^b^	65.3 ± 4.3 ^b^	40.9 ± 3.1 ^a^	72.7 ± 3.3 ^b^	50.4 ± 3.1 ^a^*
Body fat content (%)	8.1 ± 0.3 ^a^	13.7 ± 0.4 ^b^	12.9 ± 0.6 ^b^	9.3 ± 0.6 ^a^	14.4 ± 0.6 ^b^	13.4 ± 0.5 ^b^
iWAT (%)	0.9 ± 0.1 ^a^	1.7 ± 0.1 ^b^	1.8 ± 0.1 ^b^	1.3 ± 0.1 ^a^*	1.6 ± 0.2 ^ab^	2.1 ± 0.2 ^b^
Acc. food intake ^1^ (kJ)	1239.5± 36.5 ^a^	2598.3 ± 80.5 ^b^	2696.9 ± 36.6 ^b^	1226.5 ± 40.1 ^a^	2793.0 ± 132.6 ^b^	2513.5 ± 71.1 ^b^*
Food intake ^2^ (kJ/day)	338.1 ± 18.2 ^a^	592.0 ± 12.7 ^b^	576.1 ± 32.5 ^b^	308.1 ± 4.9 ^a^	791.4 ± 77.7 ^b^*	578.1 ± 20.3 ^c^

Data represent mean ± SEM (*n* = 12–16). ^a,b,c^ represent significant differences among same-ZT groups using one-way ANOVA (*p* < 0.05) followed by Bonferroni post-hoc test. * effect of ZT treatment, comparison between two groups with the same diet and supplementation but different ZT treatment determined by Student’s *t*-test. ^1^ Accumulated food intake during treatment period; ^2^ final food intake calculated at week 9. Abbreviations: ZT0, treatment administration at the beginning of the light phase; ZT12, treatment administration at the beginning of the dark phase; STD-VH, standard diet group supplemented with vehicle; CAF-VH, cafeteria diet group supplemented with vehicle; CAF-GSPE, cafeteria diet group supplemented with grape-seed procyanidin extract; iWAT, inguinal white adipose tissue; Acc, accumulated.

**Table 2 nutrients-14-02246-t002:** Rhythm variables of main metabolic parameters in serum of rats fed standard or cafeteria diet treated with vehicle or GSPE at ZT0 or ZT12.

	Group	ZT0 Glucose	ZT0 TAG	ZT0 CHOL	ZT0 NEFA	ZT12 Glucose	ZT12 TAG	ZT12 CHOL	ZT12 NEFA
Rhythmicity (*p* value)	STD-VH	NS	0.001	NS	NS	NS	0.002	0.035	NS
	CAF-VH	0.017	0.001	NS	0.048	NS	NS	NS	NS
	CAF-GSPE	NS	0.007	0.002	0.001	NS	0.019	NS	NS
Mesor (mg/dL)	STD-VH	87.19	71.38	91.37	25.49	84.87	90.71	111.98	29.98
	CAF-VH	103.64 *	191.58 *	110.03 *	29.29	117.33 *	211.44 *	116.02	30.70
	CAF-GSPE	102.73 #	163.68 #	102.03	30.59	113.53 #	228.02 #	122.68	32.91
Amplitude estimation	STD-VH	4.92	34.38	7.89	5.37	3.14	52.06	22.48	2.13
	CAF-VH	10.51	104.54 *	20.42	4.15	11.05	62.78	16.51	3.59
	CAF-GSPE	9.23	81.10	32.14 #	6.89	9.96	93.45	17.16	2.10
Acrophase (ZT)	STD-VH	NA	6.39	NA	NA	NA	5.71	5.32	NA
	CAF-VH	10.06	8.83	NA	11.83	NA	NA	NA	NA
	CAF-GSPE	NA	7.39	5.52	12.65	NA	7.04	NA	NA

Table shows the presence of rhythm, in a 24 h period, of each biochemical parameter with a *p* value (*p* < 0.05 indicates significant rhythm), circadian mean concentration (mesor), amplitude, and acrophase in ZT (time of highest concentration within the day) followed by comparisons between two groups using CircaCompare (*p* value < 0.05). * indicates significant difference between CAF-VH and STD-VH; # indicates significant difference between CAF-GSPE and STD-VH. NA indicates not-available acrophase values in non-rhythmic metabolites, and NS indicates not significant. Abbreviations: ZT0, treatment administration at the beginning of the light phase; ZT12, treatment administration at the beginning of the dark phase; STD-VH, standard diet group supplemented with vehicle; CAF-VH, cafeteria diet group supplemented with vehicle; CAF-GSPE, cafeteria diet group supplemented with grape-seed procyanidin extract; TAG, tryacilglicerides; CHOL, cholesterol; NEFA, non-esterified fatty acids.

**Table 3 nutrients-14-02246-t003:** Daily (24-h mean) concentrations (μmol/g) of individual lipophilic metabolites identified in iWAT.

	ZT0 STD-VH	ZT0 CAF-VH	ZT0 CAF-GSPE	ZT12 STD-VH	ZT12 CAF-VH	ZT12 CAF-GSPE
Omega-3	189.7 ± 8.1 ^a^	137.0 ± 6.1 ^b^	156.3 ± 9.3 ^b^	210.0 ± 10.7 ^a^	145.5 ± 5.5 ^b^	161.5 ± 8.8 ^b^
Oleic Acid (18:1)	1641.2 ± 106.7 ^a^	2402.5 ± 109.3 ^b^	2285.6 ± 125.7 ^b^	1553.2 ± 128.7 ^a^	2457.2 ± 105.8 ^b^	2196.3 ± 119.7 ^b^
Linoleic Acid (18:2)	528.8 ± 24.8 ^a^	383.0 ± 19.8 ^b^	361.5 ± 18.4 ^b^	456.5 ± 38.6 ^a^	354.0 ± 20.3 ^b^	352.2 ± 23.4 ^b^
PUFAs	523.2 ± 26.9 ^a^	269.2 ± 12.4 ^b^	276.6 ± 24.0 ^b^	500.9 ± 21.8 ^a^	252.1 ± 9.7 ^b^	294.5 ± 16.5 ^b^
Total Fatty Acid	2905.8 ± 122.9 ^a^	3262.3 ± 148.9 ^b^	3121.7 ± 111.8 ^ab^	2587.7 ± 231.8 ^a^	3279.1 ± 121.9 ^b^	3108.5 ± 158.4 ^ab^
Diglycerides	3.4 ± 0.5	5.0 ± 0.4	3.9 ± 0.6	3.3 ± 0.5	4.6 ± 0.4	4.0 ± 0.4
Triglycerides	1467.5 ± 62.1	1636.0 ± 65.6	1601.1 ± 59.7	1338.9 ± 93.5 ^a^	1658.5 ± 62.9 ^b^	1557.0 ± 66.5 ^ab^
Sphingomyelin	0.3 ± 0.05	0.2 ± 0.04	0.2 ± 0.03	0.3 ± 0.06	0.2 ± 0.02	0.2 ± 0.07
Cholesterol	5.9 ± 0.4	5.5 ± 0.5	6.5 ± 0.8	5.8 ± 0.6	6.0 ± 0.4	6.0 ± 0.4
MUFAs	4265.6 ± 162.1	4271.7 ± 180.1	4167.9 ± 132.7	3941.3 ± 229.2	4263.5 ± 133.6	4077.3 ± 147.8

Data represent mean ± SEM (*n* = 12–16). ^a,b^; represent significant differences among same ZT groups using Kruskal–Wallis (*p* < 0.05) followed by Bonferroni post-hoc test. Metabolites are sorted starting from the lowest *p* value comparing the groups treated at ZT0. Abbreviations: ZT0, treatment administration at the beginning of the light phase; ZT12, treatment administration at the beginning of the dark phase; STD-VH, standard diet group supplemented with vehicle; CAF-VH, cafeteria diet group supplemented with vehicle; CAF-GSPE, cafeteria diet group supplemented with grape seed procyanidin extract.

**Table 4 nutrients-14-02246-t004:** Daily (24 h mean) concentrations (μmol/g) of individual hydrophilic metabolites identified in iWAT.

	ZT0 STD-VH	ZT0 CAF-VH	ZT0 CAF-GSPE	ZT12 STD-VH	ZT12 CAF-VH	ZT12 CAF-GSPE
Aspartate	0.11 ± 0.017 ^a^	0.04 ± 0.004 ^b^	0.05 ± 0.006 ^b^	0.08 ± 0.010	0.05 ± 0.006	0.05 ± 0.008
Lysine	0.11 ± 0.016 ^a^	0.06 ± 0.016 ^b^	0.07 ± 0.015 ^ab^	0.09 ± 0.011	0.06 ± 0.008	0.06 ± 0.010
Carnitine	0.01 ± 0.001 ^a^	0.005 ± 0.001 ^b^	0.01 ± 0.001 ^b^	0.01 ± 0.001	0.00 ± 0.000	0.01 ± 0.001
Taurine	1.05 ± 0.110 ^a^	0.68 ± 0.058 ^b^	0.78 ± 0.094 ^b^	0.97 ± 0.080	0.70 ± 0.055	0.73 ± 0.072
Propionate	0.02 ± 0.003 ^a^	0.01 ± 0.002 ^b^	0.01 ± 0.002 ^ab^	0.02 ± 0.002 ^a^	0.01 ± 0.001 ^b^	0.01 ± 0.002 ^ab^
GPC	0.11 ± 0.018 ^a^	0.06 ± 0.012 ^b^	0.07 ± 0.013 ^ab^	0.09 ± 0.012	0.08 ± 0.011	0.07 ± 0.001
Choline	0.01 ± 0.001 ^a^	0.01 ± 0.001 ^b^	0.01 ± 0.001 ^ab^	0.01 ± 0.001	0.01 ± 0.001	0.01 ± 0.001
Inosine	0.08 ± 0.011 ^a^	0.05 ± 0.006 ^b^	0.06 ± 0.007 ^ab^	0.07 ± 0.006	0.05 ± 0.005	0.05 ± 0.007
Histamine	0.07 ± 0.011 ^a^	0.04 ± 0.011 ^b^	0.04 ± 0.008 ^ab^	0.06 ± 0.008	0.03 ± 0.005	0.03 ± 0.007
Glucose	0.31 ± 0.037 ^a^	0.19 ± 0.017 ^b^	0.26 ± 0.04 ^ab^	0.27 ± 0.034	0.22 ± 0.021	0.21 ± 0.024
Glutamine	0.96 ± 0.127	0.61 ± 0.078	0.72 ± 0.116	0.85 ± 0.091	0.64 ± 0.069	0.64 ± 0.085
Lactate	1.05 ± 0.149	0.64 ± 0.089	0.74 ± 0.120	0.93 ± 0.145	0.71 ± 0.116	0.63 ± 0.100
Leucine	0.07 ± 0.011	0.04 ± 0.007	0.04 ± 0.008	0.06 ± 0.008	0.04 ± 0.005	0.04 ± 0.005
Creatine	0.15 ± 0.043	0.08 ± 0.011	0.10 ± 0.022	0.11 ± 0.022	0.07 ± 0.006	0.09 ± 0.019
Isoleucine	0.03 ± 0.005	0.02 ± 0.003	0.02 ± 0.003	0.03 ± 0.004	0.02 ± 0.002	0.02 ± 0.003
Glycogen	0.46 ± 0.156	0.17 ± 0.033	0.22 ± 0.06	0.20 ± 0.085 *	0.31 ± 0.114	0.20 ± 0.065
Glutamate	0.23 ± 0.034	0.15 ± 0.017	0.18 ± 0.022	0.21 ± 0.025	0.16 ± 0.015	0.16 ± 0.019
Succinate	0.04 ± 0.006	0.03 ± 0.003	0.03 ± 0.003	0.03 ± 0.004	0.03 ± 0.003	0.03 ± 0.005
Valine	0.05 ± 0.008	0.03 ± 0.004	0.04 ± 0.006	0.05 ± 0.006	0.04 ± 0.003	0.03 ± 0.004
Uridine	0.03 ± 0.005	0.02 ± 0.003	0.02 ± 0.004	0.03 ± 0.004	0.02 ± 0.002	0.03 ± 0.004
Alanine	0.31 ± 0.051	0.20 ± 0.029	0.23 ± 0.033	0.28 ± 0.032	0.21 ± 0.023	0.21 ± 0.027
Myo-inositol	0.59 ± 0.141	0.32 ± 0.078	0.37 ± 0.086	0.52 ± 0.083	0.37 ± 0.074	0.34 ± 0.077
Asparagine	0.03 ± 0.005	0.01 ± 0.002	0.02 ± 0.004	0.03 ± 0.005	0.02 ± 0.002	0.02 ± 0.003
ChoP	0.07 ± 0.013	0.05 ± 0.007	0.06 ± 0.008	0.07 ± 0.011	0.05 ± 0.006	0.06 ± 0.011
Glycerol	0.09 ± 0.008	0.09 ± 0.006	0.10 ± 0.009	0.10 ± 0.010	0.08 ± 0.005	0.09 ± 0.007
Phenylalanine	0.03 ± 0.006	0.02 ± 0.003	0.02 ± 0.005	0.03 ± 0.006	0.02 ± 0.002	0.02 ± 0.003
Tyrosine	0.03 ± 0.007	0.02 ± 0.004	0.02 ± 0.004	0.03 ± 0.004	0.02 ± 0.002	0.02 ± 0.003
Niacinamide	0.02 ± 0.004	0.01 ± 0.002	0.01 ± 0.003	0.02 ± 0.002	0.01 ± 0.002	0.02 ± 0.002
AMP	0.03 ± 0.005	0.03 ± 0.008	0.03 ± 0.004	0.02 ± 0.003 *	0.02 ± 0.004	0.02 ± 0.004
Uracil	0.01 ± 0.002	0.01 ± 0.002	0.01 ± 0.003	0.01 ± 0.002	0.01 ± 0.001	0.01 ± 0.002
Fumarate	0.003 ± 0.001	0.003 ± 0.001	0.002 ± 0.000	0.004 ± 0.001	0.002 ± 0.000	0.003 ± 0.001
3-OHB	0.02 ± 0.003	0.03 ± 0.003	0.03 ± 0.006	0.03 ± 0.007	0.02 ± 0.004	0.03 ± 0.005
Acetate	0.05 ± 0.006	0.05 ± 0.004	0.05 ± 0.008	0.06 ± 0.006	0.04 ± 0.003	0.05 ± 0.004
Sarcosine	0.02 ± 0.002	0.02 ± 0.002	0.02 ± 0.002	0.02 ± 0.003	0.01 ± 0.002	0.02 ± 0.005
Formate	0.01 ± 0.001	0.01 ± 0.001	0.01 ± 0.002	0.01 ± 0.001	0.01 ± 0.001	0.01 ± 0.001

Data represent mean ± SEM (*n* = 12–16). ^a,b^; represent significant differences among same ZT groups using Kruskal–Wallis (*p* < 0.05) followed by Bonferroni post-hoc test, * indicates differences between ZTS STD-VH vs ZT12 STD-VH using Mann–Whitney (*p* < 0.05). Metabolites are sorted starting from the lowest *p* value comparing the groups treated at ZT0. Abbreviations: ZT0, treatment administration at the beginning of the light phase; ZT12, treatment administration at the beginning of the dark phase; STD-VH, standard diet group supplemented with vehicle; CAF-VH, cafeteria diet group supplemented with vehicle; CAF-GSPE, cafeteria diet group supplemented with grape seed procyanidin extract; GPC, glycerophosphocholine; ChoP, phosphorylcholine; 3-OHB, 3-hydorxybutyrate.

**Table 5 nutrients-14-02246-t005:** Rhythmic metabolic pathways detected at ZT12 in iWAT.

Group	Metabolic Pathway with Diurnal Rhythm	Total	Hits	*p* Value	FDR	Impact
STD-VH	Alanine, aspartate, and glutamate metabolism	28	5	1.10 × 10^−6^	9.20 × 10^−^^5^	3.10 × 10^−^^1^
	Arginine biosynthesis	14	3	1.30 × 10^−^^4^	4.60 × 10^−^^3^	1.20 × 10^−^^1^
	D-Glutamine and D-glutamate metabolism	6	2	8.50 × 10^−^^4^	1.20 × 10^−^^2^	5.00 × 10^−^^1^
CAF-GSPE	Alanine, aspartate, and glutamate metabolism	28	4	3.80 × 10^−^^4^	1.60 × 10^−^^2^	3.40 × 10^−^^1^
	Phenylalanine, tyrosine, and tryptophan biosynthesis	4	2	9.90 × 10^−^^4^	2.80 × 10^−^^2^	1.00

Abbreviations: STD-VH-ZT12, standard diet supplemented with vehicle at the beginning of the dark phase; CAFGSPE-ZT12, cafeteria diet supplemented with grape seed procyanidin extract at the beginning of the dark phase. Hydrophilic metabolites with significant rhythmicity were analyzed for pathway enrichment using MetaboAnalyst. *p*-value < 0.05, FDR < 0.05 and impact > 0.1 was considered significant.

## Data Availability

The data presented in this study are available on request from the corresponding author. The data are not publicly available due to lack of platform to publish them.

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
