# Peer review of "Proanthocyanidins Restore the Metabolic Diurnal Rhythm of Subcutaneous White Adipose Tissue According to Time-Of-Day Consumption"

_nutrients, 2022, doi:10.3390/nu14112246_

Round 1
Reviewer 1 Report
Authors have studied the effect of GSPE consumption on rhythmic metabolites of subcutaneous WAT by administration either at the beginning of the light/rest phase or at the beginning of the dark/active phase in rats with diet-induced obesity. They have selected a good topic but the manuscript needs some improvements before its publication in Nutrients which are as follows:
1. Besides Fig. S1, the representative images of H&E staining need to be provided to evaluate GSPE administration on the histology of iWAT.
2. In Table 2, please check (p > 0.05 indicates significant rhythm); the abbreviation of NS need annotation.
3. The resolution of Fig. S2 is too low.
4. The authors need to compare their results with previous studies on the effects of GSPE or proanthocyanidins on body weight gain, body fat content, iWAT, and blood lipid.
5. The authors evaluated the effect of GSPE consumption on rhythmic metabolites of subcutaneous WAT, however, the mechanism behind need to be deeply discussed.
6. As shown in Figure.1b, GSPE supplementation at ZT12 restored the rhythm of 12 of 16 metabolites. In addition, GSPE administration at ZT12 also resulted in another 13 rhythmic metabolites, what about the roles of these 13 metabolites in metabolic diurnal rhythm of iWAT?
7. Authors have summarized the metabolic network of identified rhythmic metabolites in iWAT in Figure.4, however, the effects of rhythmic metabolites on changes in concentrations of omega-3, oleic acid, linoleic acid, PUFAs and MUFAs remains unclear.
Reviewer 2 Report
The manuscript entitled “Proanthocyanidins restore the metabolic diurnal rhythm of subcutaneous white adipose tissue according to time-of-day consumption”, authored by Marina Colom-Pellicer, deals with the investigation of the effects on white adipose tissue rhythmicity in a time-dependent manner after the consumption of grape seed proanthocyanidin extracts. The study presented by the authors appears to be very innovative from several points of view. First, the authors investigated the potential use of an extract derived from an agro-food waste as a supplement to the animal diet, supporting their use under a circular economy perspective. Second, the data reported in the manuscript show a real benefit on white adipose tissue from a time perspective, which completes the picture of the current state of the art. The manuscript turns out to be clear in all parts, well written, and discussed by comparing the obtain results with appropriate bibliographic references. Below, I would like to suggest a number of changes that could increase the quality of this manuscript after its publication.
AFFILIATION SECTION: Affiliations lack the acronyms of each author after reporting the email address. The same acronyms should be used for the contributions section at the end of the manuscript.
KEYWORDS SECTION: regarding the keywords, they are a useful tool to help indexers and search engines to find relevant papers of interest. If scientific search engines (such as PubMed, Scopus, Google Scholar, etc) can find a potential manuscript by the use of words contained in both title, abstract, and keywords. Consequently, readers will be able to find it too thank this words. An easier search of the manuscript allows to increase the number of people reading your manuscript after publication and, then, to obtain more citations. Consequently, keywords should be words preferably not contained in the title or abstract. This short explanation is to suggest that authors introduce as many keywords as they can, and replace those words that are already present at least in the title with new keywords properly related to the reviewed manuscript.
INTRODUCTION SECTION: regarding the first paragraph concerning information related to anatomy and physiology of adipose tissue, I have no particular points to make. The authors have been comprehensive and have best described the current state of the art. However, although the experimental model used by the authors is animal, the real "star" of this manuscript are proanthocyanidins (PACs). As briefly mentioned by the authors, PACs are important polyphenolic compounds derived from the condensation of flavan-3-ols subunits. PACs have a truly limited distribution in the plant kingdom, and not all plant species have the capacity to produce these particular polyphenolic compounds, as they lack the genes necessary for transcription of the enzymes that convert other polyphenols in flavan-3-ols or (more expected) miss the pathway that allows the condensation of flavan-3-ols into PACs (10.3390/antiox10081229; 10.1021/acs.jafc.8b02950). In addition, the authors should also specify that within the PACs-producing plants, some selectively produce A-type PACs (double bond: C4-C8 and C2-O7), while others produce B-type PACs (single bond: C4-C8 or C4-C6). The structural difference of PACs determines and influences their bioactivity. In particular, the plant material used by the authors turns out to be one of the waste products with the highest content of B-type PACs (10.3390/antiox10081229). Consequently, the authors should introduce some information regarding both the distribution in plant kingdom and differences in chemical structure, highlighting in discussion section that the observed effects should be ascribed to the presence of B-type PACs, e not A-type.
RESULTS SECTION: Footnotes at the bottom of the table should be avoided. Authors should report the information describing the experimental conditions in the caption of the table, and use the footnotes only for explanation of acronyms and symbols.
DISCUSSION: The discussion section appears to be too long to be presented as a single paragraph. For ease of reading and comprehension, perhaps it would be appropriate to divide it into smaller subsections, leaving lines 378- 386 as a brief introduction of subsequent discussion sections.
CONCLUSION: This section should be better argued, more fully illustrating and reporting the obtained results.
